# Guanidinocalix[5]arene for sensitive fluorescence detection and magnetic removal of perfluorinated pollutants

Zhe Zheng[1], Huijuan Yu[2], Wen-Chao Geng[1], Xin-Yue Hu[1], Yu-Ying Wang[1], Zhihao Li[1], Yuefei Wang[2] & Dong-Sheng Guo [1]*

Perfluorinated alkyl substances, such as perfluorooctane sulfonate (PFOS) and perfluorooctanoic acid (PFOA), are toxic materials that are known to globally contaminate water, air, and soil resources. Strategies for the simultaneous detection and removal of these compounds are desired to address this emerging health and environmental issue. Herein, we develop a type of guanidinocalix[5]arene that can selectively and strongly bind to PFOS and PFOA, which we use to demonstrate the sensitive and quantitative detection of these compounds in contaminated water through a fluorescent indicator displacement assay. Moreover, by co-assembling iron oxide nanoparticle with the amphiphilic guanidinocalix[5]arene, we are able to use simple magnetic absorption and filtration to efficiently remove PFOS and PFOA from contaminated water. This supramolecular approach that uses both molecular recognition and self-assembly of macrocyclic amphiphiles is promising for the detection and remediation of water pollution.

[1] College of Chemistry, State Key Laboratory of Elemento-Organic Chemistry, Key Laboratory of Functional Polymer Materials (Ministry of Education), Nankai University, Tianjin 300071, China. [2] Institute of Traditional Chinese Medicine, Tianjin University of Traditional Chinese Medicine, Tianjin 301617, China. *email: dshguo@nankai.edu.cn

Water pollution is a serious threat to the health of organisms worldwide and is widely regarded as a major environmental issue[1]. The contamination of surface and ground water[2] by perfluorinated alkyl substances has particularly emerged as an environmental crisis impacting hundreds of millions of people[3,4] due to the increasing use of these compounds in the production of fluoropolymers[5], stain guard products[6], and fire-fighting foams[7]. The most common perfluorinated alkyl pollutants are perfluorooctane sulfonate (PFOS) and perfluorooctanoic acid (PFOA) (Fig. 1), which have been found in water worldwide, including the polar zones[8]. PFOS and PFOA can bind to proteins[9] and then deposit within the body, resulting in undesirable effects[10], including kidney[11] and liver[12] damage, thyroid disease[13], immunotoxicity[14], reproductive toxicity[15], and cancer[16]. Moreover, these pollutants are highly stable as a result of the thermodynamic stability of the C–F bonds[17] and demonstrate significant bioaccumulation, making them a persistent chemical threat to the environment[18].

According to the U.S. Environmental Protection Agency, the health advisory level for the combined concentration of PFOA and PFOS in drinking water is $70\,ng\,L^{-1}$ [3]. Nevertheless, the levels of them in drinking water typically exceed that threshold in communities near industrial areas, airports, and military facilities[3,19,20]. As a result, there is an increasing need for rapid and sensitive techniques for the detection of PFOS and PFOA in contaminated water, as well as efficient remediation methods.

Typical non-labeled detection methods of PFOS and PFOA, include gas chromatography with electron capture detection[21] and chromatography-mass spectrometry[3]. However, the long analysis time and requirement of expensive instrumentation hinders the application of these techniques in high-throughput screening of environmental samples. There have been limited studies on the development of simple, inexpensive, and sensitive optical methods (e.g., via fluorescence or colorimetric changes) for the detection of PFOS[22] and PFOA[23,24]. Furthermore, the lack of a specific receptor design can limit the sensitivity and selectivity of these optical detection methods[25].

Degrading PFOA and PFOS is difficult and will produce new toxic byproducts[26]. Therefore, absorption may be the most suitable technique for purifying PFOS and PFOA from contaminated water[27]. Typical absorbents are made of activated carbon[28], carbon nanotubes[29], resins[28], polymers[30], mineral

materials, biomaterials, and molecularly imprinted polymers[31]. However, there are few absorbents specifically designed for perfluorinated alkyl substances[32], and no absorption system has been integrated with real-time detection through simple optical techniques.

Supramolecular chemistry represents an elegant approach to construct multifunctional materials for challenging applications[33,34], with the virtues of molecular recognition and self-assembly. Macrocyclic hosts, such as crown ether[35], cyclodextrin[36], calixarene[37–40], pillararene[41,42], cucurbituril[43,44], and others[45,46], are families of well-studied artificial receptors with a cavity that can be selective for the recognition of particular guests. Efficient molecular recognition by macrocycles in aqueous media has been demonstrated for various applications, including the detection[44,47] and absorption[48,49] of pollutants. One representative example is β-cyclodextrin[50], which displays a binding affinity of $\sim10^4\,M^{-1}$ for PFOS and PFOA[51], enabling a β-cyclodextrin polymer network to efficiently sequester PFOA[27,52]. Exploring artificial receptors with extraordinarily high affinities to PFOS and PFOA is crucial for both the detection and absorption of these compounds, as stronger binding will result in higher sensitivity and absorption efficiency.

In this work, we report the nanomolar binding of two different guanidinocalix[5]arenes (GC5A-6C and GC5A-12C, which feature 6 and 12 carbon atoms in each alkyl chains at the lower rim of the macrocycles, respectively) (Fig. 1) towards PFOS and PFOA. Based on molecular recognition, we achieve sensitive fluorescence detection of PFOA and PFOS using an indicator displacement assay (IDA) with fluorescein (Fl) as the reporter dye (Fig. 1). Moreover, efficient magnetic absorption of PFOA and PFOS is achieved by loading magnetic iron oxide nanoparticle (MNP) into the GC5A-12C assembly.

## Results

**Complexation studied by theoretical calculations and NMR.** Both PFOS and PFOA possess two potential binding sites: a head group composed of sulfonate (PFOS) or carboxylic acid (PFOA) and a C–F chain tail. Based on these structural features, we explored GC5A-6C as a candidate receptor mainly due to its complementary size and shape compared to the chain structures of PFOS and PFOA, as well as its salt-bridge interactions. GC5A-6C was synthesized and purified according to our previous method[53]. Geometry optimizations of the GC5A-6C•PFOA and GC5A-6C•PFOS complexes were performed using the B3LYP-D3/6–31G(d)/SMD(water) method (Supplementary Note 1)[54–57], in which we found that both complexes feature a threading geometry (Fig. 2a and Supplementary Fig. 1).

To gain further understanding of the molecular recognition behavior, we calculated and mapped the molecular electrostatic potential (MEP)[59] on the molecular van der Waals surfaces of GC5A-6C and PFOS (Fig. 2a). GC5A-6C is electron-deficient, especially at its upper rim, and PFOS is electron-rich, particularly at its sulfonate head group. The optimized binding geometry is reasonable since molecules tend to approach each other in a complementary manner of the MEP. Furthermore, independent gradient model analysis[58] (Fig. 2b) reveals the strong N–H⋯O hydrogen bonds (blue areas in the isosurfaces) between the guanidinium groups of GC5A-6C and the head group of PFOS. The green areas in the isosurfaces indicate the existence of (1) weak C–H⋯F–C hydrogen bonds between alkyl chains in the GC5A-6C and the fluorocarbon chain in PFOS[60], and (2) van der Waals interactions of the aromatic rings and oxygen atoms in GC5A-6C with the fluorocarbon chain in PFOS. Coloring the atoms of GC5A-6C according to their contribution to the

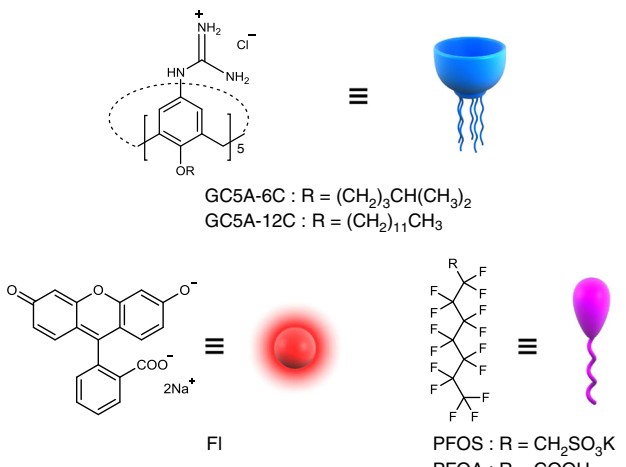

**Fig. 1 Chemical structures of hosts and guests.** Chemical structures of the employed calixarene hosts (GC5A-6C and GC5A-12C), the Fl reporter dye, and the perfluorinated pollutants (PFOS and PFOA) in this work.

GC5A-6C : R = $(CH_2)_3CH(CH_3)_2$
GC5A-12C : R = $(CH_2)_{11}CH_3$

Fl

PFOS : R = $CH_2SO_3K$
PFOA : R = COOH

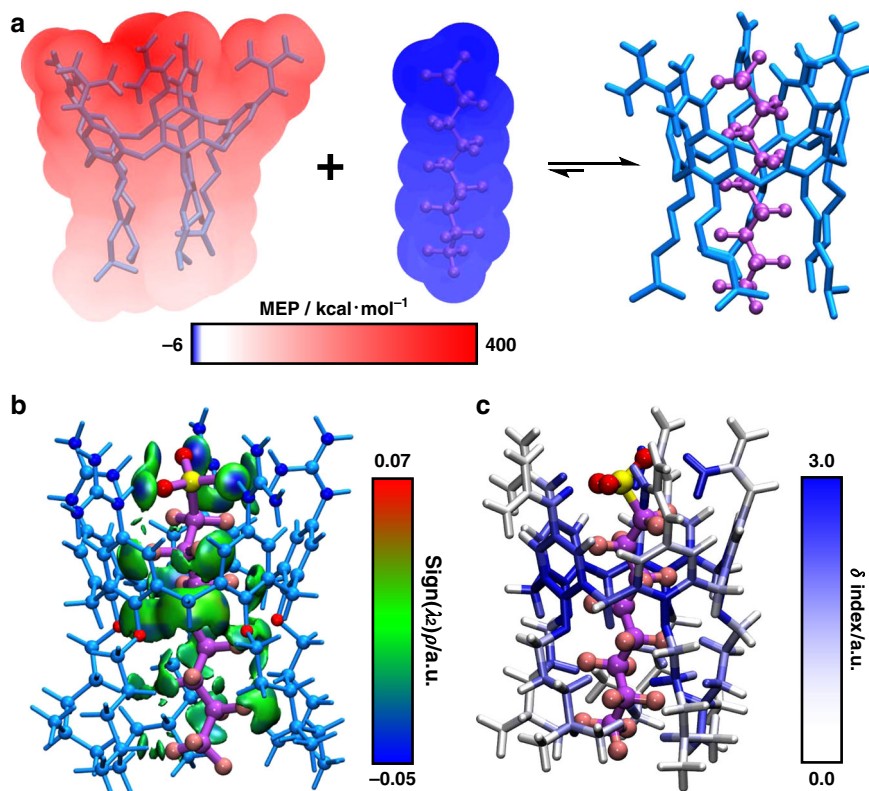

**Fig. 2 Binding geometry of GC5A-6C and PFOS. a** The optimized binding geometry of the GC5A-6C•PFOS complex (right) at the B3LYP-D3/6–31G(d)/ SMD(water) level of theory and the MEP-mapped molecular van der Waals surfaces of GC5A-6C (left) and PFOS (middle). **b** $\delta g^{inter} = 0.01$ a.u. isosurfaces colored by the sign of $(\lambda_2)\rho$ for the GC5A-6C•PFOS complex (the meaning of $\delta g^{inter}$ and sign$(\lambda_2)\rho$ were in ref. [58]). Blue indicates strong attraction, while red indicates strong repulsion. **c** The atoms of GC5A-6C colored according to their contributions to the binding with PFOS. White indicates no contribution to the complexation, and blue indicates the largest relative contribution.

host-guest complexation clearly shows that the main contributions derive from the guanidinium groups and aromatic rings, though the alkyl chains also provide weak interactions (Fig. 2c). As a result, the synergistic effect of these different interactions contributes to the strong binding between GC5A-6C and PFOS or PFOA as desired. We verified the complexation of PFOA with GC5A-6C by $^{19}$F nuclear magnetic resonance (NMR) experiments in CD$_3$OD. The significant upfield shift in the fluorine nuclei link to C$_\alpha$ was observed (Supplementary Fig. 2), which may be caused by the intermolecular N–H···F hydrogen bonds between guanidinium groups of GC5A-6C and fluorine atoms link to C$_\alpha$ of PFOA when PFOA was encapsulated into the cavity of GC5A-6C (Supplementary Fig. 1).

**Sensitive and selective detection of PFOS and PFOA.** We further determined the binding affinities of GC5A-6C to PFOS and PFOA using an IDA (Fig. 3a), in which a fluorescent indicator first reversibly binds to the receptor (Supplementary Fig. 3). Then, an analyte is added into the solution, displacing the indicator from the cavity of the receptor, which changes the optical signal[61]. By employing GC5A-6C•Fl as the reporter pair ($K_a = 5.0 \times 10^6$ M$^{-1}$)[53], we obtained binding affinities of $(3.5 \pm 1.0) \times 10^7$ M$^{-1}$ for PFOS (Fig. 3b) and $(1.7 \pm 0.3) \times 10^7$ M$^{-1}$ for PFOA (Fig. 3c). The binding affinities are about three orders of magnitude higher than those of previously reported supramolecular hosts toward PFOS, which are around $10^4$ M$^{-1}$ (Supplementary Table 3).

The principle of the IDA allows for fluorescence "switch-on" sensing of PFOS and PFOA using the GC5A-6C•Fl reporter pair. The fluorescence increase linearly with increasing PFOS (Fig. 4a) and PFOA (Fig. 4b) concentrations, respectively. Based on these results, we calculated the limit of detection (LOD) values as $21.4 \pm 0.4$ nM ($11.3 \pm 0.2$ µg L$^{-1}$) for PFOS and $26.4 \pm 0.2$ nM ($10.9 \pm 0.1$ µg L$^{-1}$) for PFOA by a $3\sigma$/slope method[62]. The sensitivity of this assay to PFOS and PFOA is superior to or comparable with those of gas chromatography and high-performance liquid chromatography techniques[63], indicating that this facile and sensitive fluorescence assay has potential for rapid detection of heavy contamination by PFOS and PFOA[21,63]. For example, the PFOA concentration in drinking water near a fluorochemical facility in Washington, West Virginia, USA, is as high as 13.3 µg L$^{-1}$, which is 190-fold greater than the health advisory level (70 ng L$^{-1}$) recommended by the U.S. Environmental Protection Agency[3,64]. Given the sensitivity of our technique, it should be possible to quickly detect PFOS and PFOA in the drinking water of such contaminated regions.

We also evaluated the detection selectivity of GC5A-6C for PFOA and PFOS. For comparison, we determined that octanesulfonic acid and octanoic acid (C–H chain surfactants) exhibit affinities of $(6.0 \pm 1.1) \times 10^4$ M$^{-1}$ (Supplementary Fig. 4) and $(7.6 \pm 0.8) \times 10^4$ M$^{-1}$ (Supplementary Fig. 5) with GC5A-6C, respectively. For these C–H chain surfactants, the hydrogen atoms feature a 23% smaller van der Waals radius compared to fluorine[44], which leads to smaller molecular volumes. As a result,

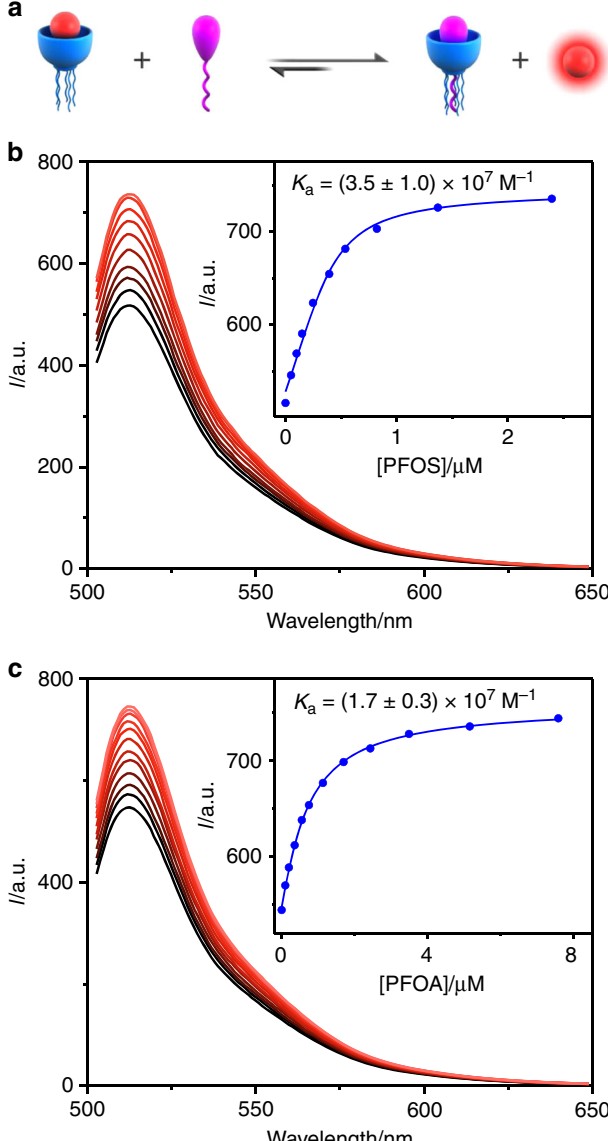

**Fig. 3 Illustration of the IDA principle and fluorescence titrations.**
**a** Illustration of the IDA principle. Competitive titrations of GC5A-6C•Fl (0.4/0.5 μM) with **b** PFOS (up to 2.4 μM) and **c** PFOA (up to 7.6 μM). (Inset) The associated titration curves of **b** PFOS and **c** PFOA fit according to a 1:1 competitive binding model. All experiments were performed in HEPES buffer at 25 °C, $\lambda_{ex}$ = 500 nm, and $\lambda_{em}$ = 513 nm. Data represent mean ± s.d. ($n$ = 3 independent experiments).

their binding affinities are almost three orders of magnitude lower than those of PFOS and PFOA. Furthermore, the addition of common anionic/cationic surfactants, perfluorohexane and anions caused no significant enhancement in the fluorescence intensity of the GC5A-6C•Fl complex (Fig. 5). Salt concentrations are considered to be orders of magnitude higher than PFOS and PFOA in contaminated water. Even 1000-fold excess salts (NaCl, KCl, and MgCl$_2$) resulted in much smaller fluorescence recovery of the GC5A-6C•Fl reporter pair than PFOS and PFOA (Supplementary Fig. 6). Considering that other unknown pollutants in real highly contaminated water might interfere with the detection selectivity of GC5A-6C, we collected the waste water samples from the manufacturing facility (Cangzhou, Hebei

Province, China), and freeze-dried these highly contaminated water samples. The addition of the obtained powder with mass concentration more than ten times higher than PFOS and PFOA also caused no significant enhancement in the fluorescence intensity, which validated GC5A-6C could bind preferentially to PFOS and PFOA over other pollutants in the real world. Moreover, the selectivity to various interferences in Fig. 5 were established in the real highly contaminated water (Supplementary Fig. 7).

To evaluate the applicability and reliability of the proposed method and considering interfering species commonly found in environmental water samples from different sources, we applied the assay to detect PFOS and PFOA in tap water and lake water samples. The water samples were obtained from tap water and Mati Lake (Nankai University, Tianjin, China), which were immediately filtered through 0.45 μm micropore membranes to remove insoluble particles and stored in brown glass bottles as blank samples. Then, we performed the IDA of PFOS and PFOA in the tap and lake water samples containing variable concentrations of the compounds. Despite the existence of various interfering species in these water samples, linear increases in the fluorescence of the GC5A-6C•Fl complex were observed as we increased the concentrations of PFOS and PFOA (Fig. 4c–f). These linear relationships allowed us to establish calibration curves of the intensity of fluorescence to determine the unknown concentrations of PFOS and PFOA in the tap and lake water samples. The LOD values for PFOS were 30.9 ± 0.1 nM (16.6 ± 0.1 μg L$^{-1}$) in the tap water and 120.8 ± 0.5 nM (65.0 ± 0.3 μg L$^{-1}$) in Mati Lake. Meanwhile, the LOD values for PFOA were 39.1 ± 2.8 nM (16.2 ± 1.2 μg L$^{-1}$) in the tap water and 120.3 ± 2.0 nM (49.8 ± 0.8 μg L$^{-1}$) in Mati Lake.

According to the LOD values of the water samples, our method can be directly applied for the detection of PFOS and PFOA in heavily contaminated sources, such as communities near industrial areas[64], airports[3], and military facilities[3], in which the total concentrations of PFOS and PFOA may reach 1174 μg L$^{-1}$[65]. For measuring contamination in regular drinking water with lower PFOS and PFOA concentrations, we can employ our assay after preconcentrating the water sample with a solid-phase extraction. Aliquots of blank tap water samples from different regions and Haihe River water samples and those spiked with 50 ng L$^{-1}$ of PFOS or PFOA were extracted with HLB cartridges (Supplementary Note 2). We then performed the displacement assay using the GC5A-6C•Fl reporter pair and obtained the concentrations of PFOS and PFOA from the established calibration curves. As shown in Supplementary Table 4, the experimental results showed that no PFOS or PFOA was detected in any of the blank tap water samples and Haihe River water samples. The standard addition recoveries of those water samples are in the range of 90.3–102.7%. These values validated our assay, demonstrating its precision and accuracy for the detection of PFOS and PFOA in contaminated drinking water.

**Real-time/on-site scanometric monitoring of PFOS and PFOA.** To make the assay more available to daily use, the obvious fluorescence changes of GC5A-6C•Fl reporter pair with various concentrations of PFOS and PFOA were further applied as a real-time/on-site visual detection mode by using a smartphone with easy-to-access color scanning application (app). The green color intensities ($G$ values) of the fluorescent images can be directly scanned from the app (Fig. 6 and Supplementary Note 3). According to the $G$ values, calibration curves can be set up with increasing PFOS and PFOA concentrations, respectively.

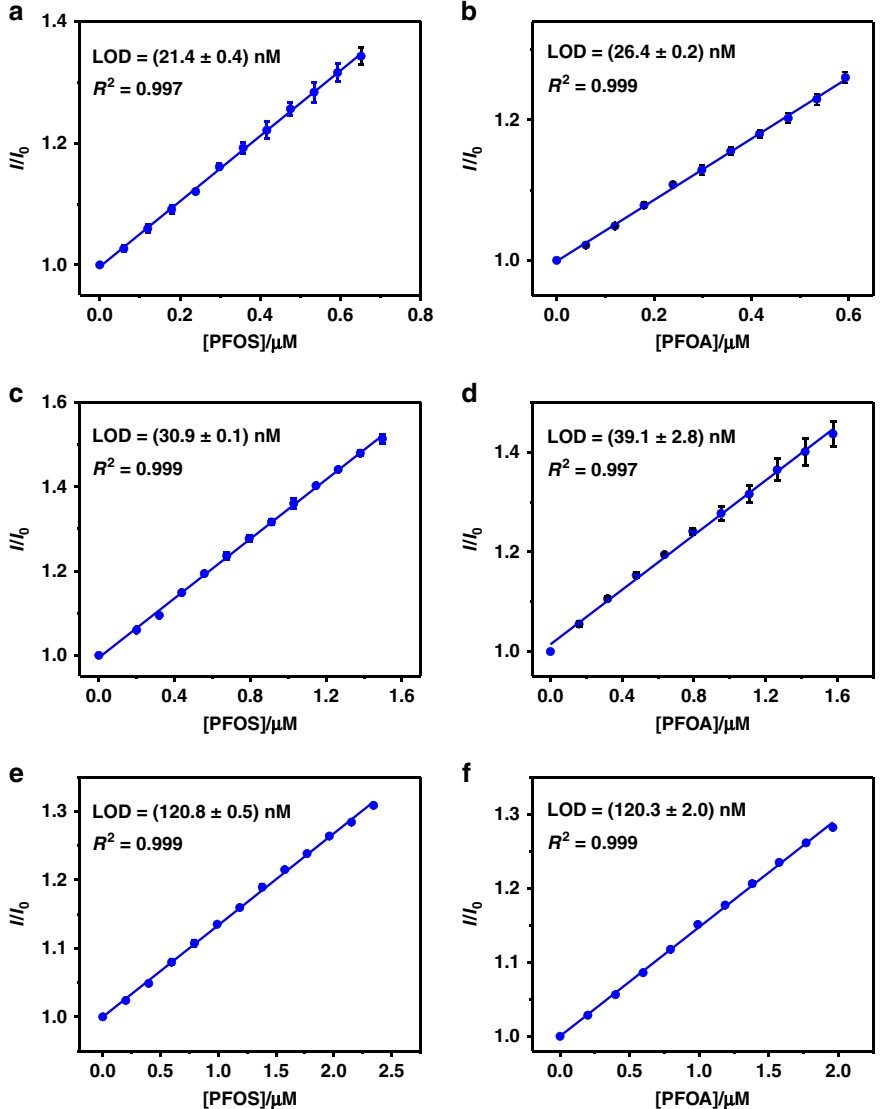

**Fig. 4 Calibration curves and LOD values of PFOS and PFOA in different media.** Plots of $I/I_0$ against PFOS (left) and PFOA (right) concentrations in **a**, **b** HEPES buffer, **c**, **d** tap water, and **e**, **f** water of Mati Lake. $I$ and $I_0$ are the intensities of fluorescence of the GC5A-6C•Fl (0.8/1.0 μM for **a**, **b**; 3.0/1.0 μM for **c**, **d**; and 10.0/1.0 μM for (**e**, **f**) reporter pair with and without the guest molecules, respectively. All experiments were performed at 25 °C, $\lambda_{ex} = 500$ nm, and $\lambda_{em} = 513$ nm. Error bars represent mean ± s.d. ($n = 3$ independent experiments).

**The absorption and magnetic separation of PFOS and PFOA.** To add absorption capability to this sensing system, we constructed a hybrid material composed of amphiphilic calixarene nanoparticle (Fig. 7). The pegylated GC5A-12C nanoparticle[66] was generated by co-assembling GC5A-12C and 4-(dodecyloxy) benzamido-terminated methoxy poly(ethyleneglycol) (PEG-12C) (Supplementary Fig. 8) at a 2:1 molar ratio (Supplementary Note 4). PEG-doping was implemented to enhance the water-solubility and mechanical stability of the material to prevent coagulation and settling[67]. We determined the binding constants for the pegylated GC5A-12C nanoparticle for PFOS and PFOA to be $(1.3 \pm 0.3) \times 10^7$ M$^{-1}$ and $(4.8 \pm 0.4) \times 10^6$ M$^{-1}$, respectively (Supplementary Figs. 9–11). The binding strength of the pegylated GC5A-12C nanoparticle is comparable with that of GC5A-6C and should be suitable for absorption applications. Furthermore, we obtained hybrid nanoparticle (MNP@GC5A-12C) by encapsulating hydrophobic MNP into the hydrophobic domain of the GC5A-12C nanoparticle during preparation (Supplementary

Note 5). Dynamic light scattering measurements revealed the MNP@GC5A-12C have a hydrated diameter of 213 ± 3 nm (Supplementary Fig. 12).

Al (III) phthalocyanine chloride tetrasulfonic acid (AlPcS$_4$, Supplementary Fig. 8) was chosen as a model dye to explore the absorption ability of MNP@GC5A-12C. MNP@GC5A-12C was dispersed into the solution of AlPcS$_4$ and then isolated with an external magnetic field for 1 h. Subsequently, the supernatant was filtered through a mixed cellulose esters film (Millipore, 0.025 μm) and collected for ultraviolet–visual (UV–vis) experiments. Negligible absorbance of AlPcS$_4$ was observed after absorption (Fig. 8a), indicating the complete removal of AlPcS$_4$ by MNP@GC5A-12C. As a control experiment, filtration without MNP@GC5A-12C resulted in barely any effect on the concentration of the AlPcS$_4$ in solution (Supplementary Fig. 13).

We further applied MNP@GC5A-12C to absorb PFOS and PFOA. The quantifications of PFOS and PFOA were performed by means of ultra-performance liquid chromatography–electrospray

ionization–tandem mass spectrometry (UPLC–ESI–MS/MS). The calibration curves were set up and quantitative parameters were evaluated (Supplementary Note 6). The PFOS and PFOA absorption

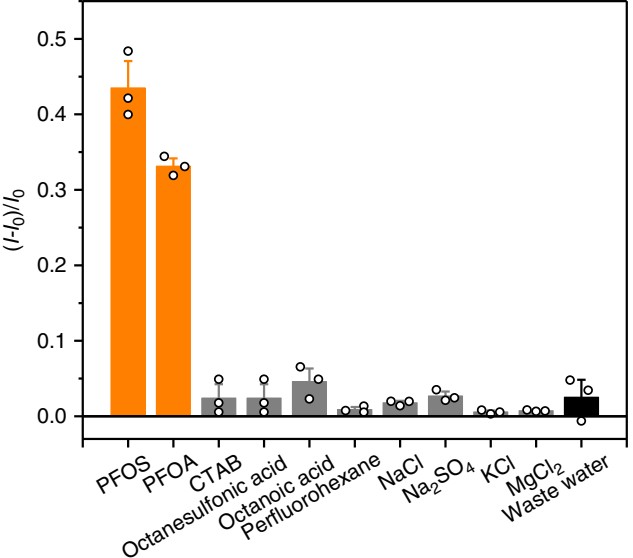

**Fig. 5 Selective detection of PFOS and PFOA.** Fluorescence responses of GC5A-6C•Fl (0.8/1.0 μM) after adding PFOS, PFOA, and interfering species (0.8 μM for small species and 5 μg mL$^{-1}$ for waste water samples). $I$ and $I_0$ are the intensities of fluorescence of the GC5A-6C•Fl reporter pair with and without the guest molecules, respectively. All experiments were performed in HEPES buffer at 25 °C, $\lambda_{ex} = 500$ nm, and $\lambda_{em} = 513$ nm. The CTAB is hexadecyltrimethylammonium bromide. Error bars represent mean ± s.d. ($n = 3$ independent experiments).

efficiencies of each samples were characterized at $[PFOS]_0 = [PFOA]_0 = 1000$ ng mL$^{-1}$. After the removal procedure, the solutions were pre-concentrated to accurately determine PFOS and PFOA concentrations by the UPLC–ESI–MS/MS. There were only $(0.43 \pm 0.07)\%$ residual PFOS and $(1.53 \pm 0.04)\%$ residual PFOA, respectively (Fig. 8b). The regeneration of the present supramolecular material is considered to be feasible owing to its response to specific organic solvent[68]. In DMSO, GC5A-12C neither formed amphiphilic aggregates indicated by very low scattering intensity (Supplementary Fig. 14), nor complexed with PFOS revealed by no change in chemical shifts of $^{19}$F NMR spectra of PFOS (Supplementary Fig. 15). Therefore, we envisage that regeneration of the supramolecular material could be achieved by using routine purification methods in organic synthesis such as column chromatography.

## Discussion

In conclusion, our artificial GC5A-6C receptor successfully encapsulated PFOA and PFOS with nanomolar affinity in aqueous media. Taking advantage of the strong recognition and supramolecular assembly, we achieved not only sensitive and quantitative detection of PFOA and PFOS in tap and lake water through the fluorescent IDA strategy, but also the efficient removal of them by the hybrid MNP@GC5A-12C nanoparticle via a simple magnetic absorption and filtration procedure. These results will facilitate the development of detection and absorption methods for PFOA and PFOS. Although the present LOD and removal efficiency cannot reach the health advisory level in drinking water, the proposed supramolecular approach can be practically operated in heavily polluted areas, such as industrial regions, airports, and military facilities. For daily drinking water detection, we can use the solid-phase extraction for sample preconcentration. Higher removal efficiency may be achieved by applying the GC5A-12C nanoparticle as solid-

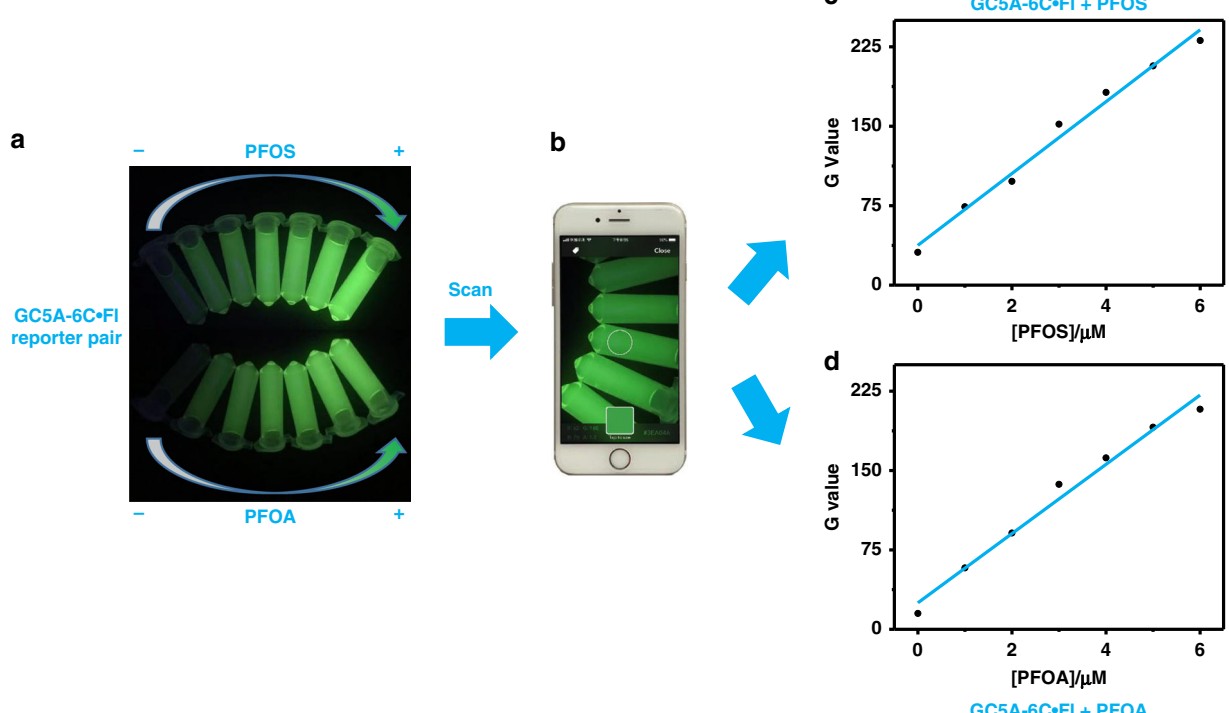

**Fig. 6 Real-time/on-site scanometric monitoring of PFOS and PFOA. a** The images of the GC5A-6C•Fl reporter pair (8.0/10.0 μM) with various concentrations of PFOS and PFOA (up to 6.0 μM) taken by an iPhone 7. **b** The images recorded by iPhone 7 with a color-scanning app. Plots of $G$ values against PFOS (**c**) and PFOA (**d**) concentrations in HEPES buffer at 25 °C. $G$ values are green color intensities directly scanned from the color scanning app.

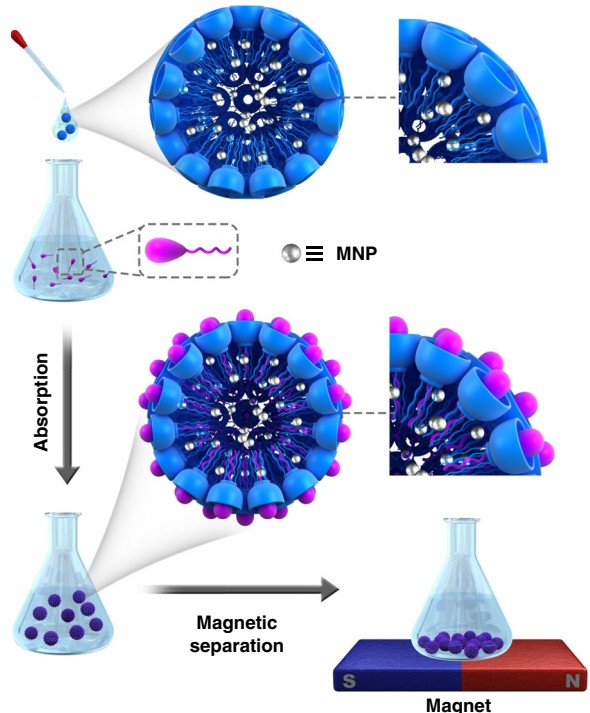

**Fig. 7 Schematic illustration of PFOS and PFOA absorption and magnetic separation.** Illustration of the absorption and magnetic separation procedure of PFOS and PFOA by the MNP@GC5A-12C nanoparticle.

phase extraction absorbents. This work made full use of the molecular recognition and self-assembly of artificial receptors, offering a promising strategy for the detection and remediation of water pollution.

## Methods

**Chemicals**. All the reagents and solvents were commercially available and used as received unless otherwise specified purification. Ammonium acetate and 2,2,2-tri-fluoroethanol were purchased from Sigma-Aldrich. Fl was purchased from Tokyo Chemical Industry. Al (III) phthalocyanine chloride tetrasulfonic acid (AlPcS$_4$) was purchased from Frontier Scientific. PFOA, PFOS, hexadecyltrimethylammonium bromide, octanoic acid, octanesulfonic acid, and perfluorohexane were purchased from Energy Chemical. Iron oxide nanoparticle stabilized by oleic acid (MNP) was purchased from Ji Cang Nano Company. The waste water samples were provided by the manufacturing facility located in Cangzhou, Hebei province, China. 5,11,17,23,29-Pentaguanidinium-31,32,33,34,35-penta(4-methylpentloxy)calix[5]arene (GC5A-6C), 5,11,17,23,29-pentaguanidinium-31,32,33,34,35-penta dodecy-loxy-calix[5]arene (GC5A-12C) and 4-(dodecyloxy)benzamido-terminated methoxy poly(ethylene glycol) (PEG-12C) were synthesized according to the previous literature[53,66].

**Samples**. The 2-[4-(2-hydroxyethyl)piperazin-1-yl]ethanesulfonic acid (HEPES) buffer solution of pH 7.4 was prepared by dissolving 2.38 g of HEPES in approximate 900 mL double-distilled water. Titrate to pH 7.4 at the lab tempera-ture of 25 °C with NaOH and make up volume to 1000 mL with double-distilled water. The pH value of the buffer solution was then verified on a pH-meter calibrated with three standard buffer solutions.

**Apparatus**. $^{19}$F NMR data were recorded on a Bruker AV400 spectrometer. Steady-state fluorescence spectra were recorded in a conventional quartz cell (light path 10 mm) on a Cary Eclipse equipped with a Cary single-cell peltier accessory. UV–vis spectra were recorded in a quartz cuvette (light path 10 mm) on a Cary 100 UV–vis spectrophotometer equipped with a Cary dual cuvette peltier acces-sory. The sample solutions for dynamic light scattering measurements were examined on a laser light scattering spectrometer (NanoBrook 173plus) equipped with a digital correlator at 659 nm at a scattering angle of 90°. Quantification of PFOS and PFOA from the absorption studies were performed by means of UPLC–ESI–MS/MS (Waters, Milford, MA, USA).

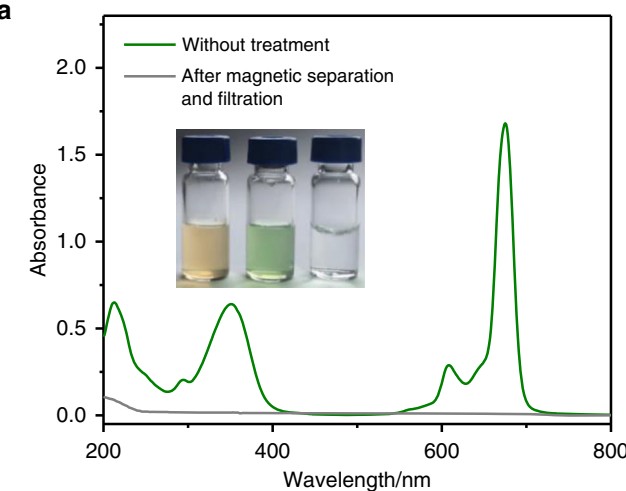

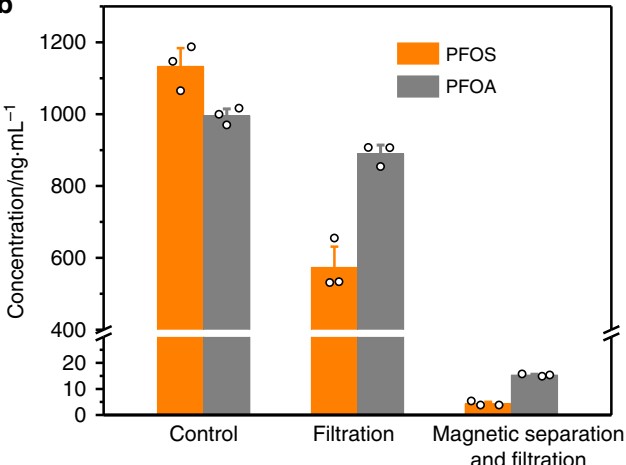

**Fig. 8 Absorption of AlPcS$_4$, PFOS, and PFOA in water samples.**
**a** Absorption spectra of AlPcS$_4$ (10 μM) without any treatment and after magnetic separation by MNP@GC5A-12C and filtration. (Inset) Photographs of MNP@GC5A-12C ([GC5A-12C] = 100 μM) (left), MNP@GC5A-12C ([GC5A-12C] = 100 μM) with AlPcS$_4$ (10 μM) (middle), and the filtrate after magnetic separation and filtration (right).
**b** Concentrations of PFOS and PFOA with only filtration, and with both magnetic separation and filtration are quantified by the UPLC–ESI-MS/MS system. Concentrations of PFOS and PFOA without magnetic separation are shown as a control. Error bars in (**b**) represent mean ± s.d. (n = 3 independent experiments).

## Data availability

The data supporting the findings of this study are available within the paper and its Supplementary Information, and from the corresponding author upon reasonable request. The source data underlying Figs. 3b, c, 4a–f, 5, 6, 8a, b, Supplementary Figs. 4, 5–7, 9–14, and Supplementary Table 4 are provided as a Source Data file.

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

## Acknowledgements

This work was supported by NSFC (21672112 and 51873090) and the Fundamental Research Funds for the Central Universities, which are gratefully acknowledged. The simulation studies were performed at the LvLiang Cloud Computing Center of China, and the calculations were performed on TianHe-2, which are gratefully acknowledged. The authors also thank Tielong Li at Nankai University for providing the waste water samples and Xianrui Wang at Tianjin University of Traditional Chinese Medicine for the UPLC–ESI–MS/MS experiments.

## Author contributions

Z.Z. and D.S.G. conceived the experiments. W.C.G. synthesized the GC5A-6C and GC5A-12C. W.C.G. and Z.L. performed the DFT calculations. Z.Z., H.Y., X.Y.H. and Y.Y.W. performed all the other experiments. Z.Z., X.Y.H., Y.W. and D.S.G. contributed to writing of this paper and all authors commented on it.

## Competing interests

The authors declare no competing interests.
