## [Peer Review File · Nature Communications]

Reviewers' comments:

Reviewer #1 (Remarks to the Author):

The manuscript submitted by Zheng et al describe the synthesis of a guanidine functionalised calix[5]arene with a reporter molecule encapsulated. The calixarene can act as a receptor for PFOA and PFOS forming a stronger complex than the calixarene-reporter, thus allowing for a fluorescent displacement assay, albeit with a relatively high LOD. Furthermore, the authors functionalise the calixarene with magnetic nanoparticles thus allowing a facile separation process.

The authors rightly identify fluoruous organic POPs as a good target for water purification, but this reviewer feels the background is not fully explored. It is appreciated that this is a communication, but setting the work thoroughly in context is important. For example, the authors discuss "heavily contaminated" areas but what are the typical limits of PFOA/PFOS in these areas? Is this a real-world applicable system or just for really highly contaminated systems?

Thinking more broadly about the impact of their work on the field, whilst their system has obvious merits in the simultaneous detection and removal of PFOA, for any commercial applications one would suspect that the ligand would be too expensive and the removal of PFOA from the calixarene to regenerate the receptor-reporter complex is not described. One could also argue that using fluorescence is not as useful as optical methods for an in the field application.

Other comments:

Calixarenes are sometimes emissive in solution without any hosts. Did the authors check that their signals are coming from a host:guest complex and not just the host? I can't see this in the manuscript.

No error bars on the data presented in Fig. 4, which is essential for a calibration graph.

Fig 5, do you mean chloride rather than chlorine?

Again on Fig 5 what about metal cations as interfering species? One would assume that the concentration of Na⁺/K⁺ etc would be orders of magnitude more than PFOA?

In summary, whilst the science is interesting and could develop the thinking of others, calixarene and cyclodextrin receptors have already been reported. The authors have references wrong or missing from this list - the group of Hazendonk has published extensively on CD, the Baker group have used calixarene (not CD in the reference list) and the Allen group have used tripodal amides (Allen, J. Org. Chem., 2009, 74, 3706) to do the same supramolecular chemistry. In all these works the host:guest complexes have been characterised in much greater detail, which leads this reviewer to err on the side of rejection due to the lack of novelty.

The applicability of this approach to water remediation is still a long way off (note that this has no bearing on the scientific merits of the manuscript for the purposes of the review), but this reviewer encourages the authors to expand their characterisation data as this work does deserve to be published in some other journal.

Reviewer #2 (Remarks to the Author):

In this manuscript, authors have developed a new strategy for the direct detection and rapid removal from water of two persistent organic pollutants (PFOS and PFOA) that are currently causing increasing environmental and health concern. The detection is based on a fluorescent indicator-displacement assay that relies on the strong affinity (in the nanomolar range) of

guanidinocalix[5]arenes GC5A-6C and GC5A-12C towards these perfluorocarbon derivatives. The removal procedure takes advantage of the amphiphilic nature of GC5A-12C and the magnetic properties of iron oxide nanoparticles that, combined in the presence of PEG-12, yield hybrid nanoparticles able to absorb, magnetically separate and remove, after simple filtration, the target analytes from water samples.

The manuscript is well written and relevant literature has been thoroughly reviewed. Experiments performed and methods used are clearly described. Discussion and conclusions are sound and backed up by solid experimental evidence (DFT calculations, ¹⁹F NMR studies, DLS measurements and UPLC-ESI-MS/MS analyses). This is a nice piece of work where molecular recognition and self-assembly properties of amphiphilic macrocycles have been cleverly combined to address an issue of current concern related to water pollution and, as a result, bioaccumulation problems and risk to human health. The detection limits reported are promising and the idea of using a magnetic separation technique is interesting. The findings by Guo and colleagues are of a high standard and will, in my opinion, provide the input for new research to be carried out in the field of environmental and separation science.

Publication of this article in Nature Communications is recommended following minor revision/additions.

Revision/additions to be addressed:

- Information concerning the regeneration of the MNP@GC5A-12C nanoparticles for subsequent reuse is lacking.
- The choice of using HEPES buffered solutions probably deserves a comment.
- Has selectivity towards simple perfluorocarbons been tested?
- The type of film used for the filtration of the supernatant obtained after the magnetic absorption should be specified (line 191).
- Replace "a family" with "families" (line 61).
- Reference 72 (line 65) deals with calix[4]arenes rather than beta-cyclodextrins!
- Authors might want to consider introducing the following publication among the references cited in the introduction: doi: 10.2903/j.efsa.2018.5194
- The caption to Fig 2 in the SI needs attention. I also wondered whether there was any specific reason for the poor phasing of the ¹⁹F NMR traces shown in Fig 2 of the SI.
- Poor trace phasing is also seen in Fig 3 of the SI.

Reviewer #3 (Remarks to the Author):

The submission by Zheng and co-workers reports a new approach to binding and removing PFOS and PFOA from water. These are highly important contaminants that are of increasing interest in the last ten years. The idea of using a general hydrophobic sorbent to reduce PFOS/PFOA levels in water is well established, and could use a better coverage in the literature citations in this work. The two main new ideas in this work are the use of a macrocyclic host for binding the fluorinated pollutants, and combining the ability to detect with the ability to remove the pollutants in the same molecular framework.

The writing is very clear and the science is well presented. The binding constants, selectivity, mode of action, and ability to remove pollutants from water are all well supported by the data presented.

In order for the work in this area to be suitable for Nature Communications and attract a broad readership, it needs both a high level of scientific innovation and a significant step toward real-world application. The innovation is indeed very high. The studies with lake water are a nice addition and they bring this quite close to real-world impact. The sorbent studies fall a bit short, mainly because the UPLC-MS method that is used to quantify PFOS/PFOA after removal are not sensitive enough to determine exactly how complete the removal of PFOS is. This ability is central

to the future utility of this method, and the analysis must be possible because it is done routinely in environmental analysis labs. I ask that the same experiments be reported with accurate determination of PFOS/PFOA concentrations down at least to the government standard of 70 ng/L. The other possible shortcoming is about selectivity, which is central to innovation and to the application. This selectivity needs to be established experimentally using real highly contaminated waters. There is a significant chance that the macrocycle will find other hydrophobic pollutants to bind preferentially over PFOS/PFOA, and this would remove the proposed advantage of the macrocyclic system, and more importantly would also decrease the innovation aspect of the work relative to nonspecific sorbents that have already been reported. If this can be established it could spark others to try their own combinations of specific macrocyclic/MNP sorbents for different species.

Reply to Reviewer 1

Comments:

The manuscript submitted by Zheng et al describe the synthesis of a guanidine functionalised calix[5]arene with a reporter molecule encapsulated. The calixarene can act as a receptor for PFOA and PFOS forming a stronger complex than the calixarene-reporter, thus allowing for a fluorescent displacement assay, albeit with a relatively high LOD. Furthermore, the authors functionalise the calixarene with magnetic nanoparticles thus allowing a facile separation process.

Response: We highly appreciate the reviewer's positive comments.

The authors rightly identify fluoruous organic POPs as a good target for water purification, but this reviewer feels the background is not fully explored. It is appreciated that this is a communication, but setting the work thoroughly in context is important. For example, the authors discuss "heavily contaminated" areas but what are the typical limits of PFOA/PFOS in these areas?

Response: To our knowledge, some researchers have reported the concentrations of PFOS and PFOA in heavily contaminated areas (Wang Y, et al. Environ. Sci. Technol., 2010, 44, 8062-8067; Exner M, et al. Environ. Sci. Pollut. Res., 2006, 13, 299-307 and Saito N, et al. J. Occup. Health, 2004, 46, 49-59). We added the description about heavily contaminated areas, "*such as communities near industrial areas, airports and military facilities, in which the total concentrations of PFOS and PFOA may reach 1174 µg/L.*" in the last paragraph of Page 6 according to the reviewer's suggestion.

Is this a real-world applicable system or just for really highly contaminated systems?

Response: The strategy constructed for PFOS and PFOA detection is not only for really highly contaminated systems, but also for a real-world application.

According to the reviewer's suggestion, we applied our detecting approach in regular drinking water with lower PFOS and PFOA concentrations than the health advisory level (70 ng/L) recommended by the U.S. Environmental Protection Agency,

giving the desirable positive results. We added Supplementary Table 3 in the Supplementary Materials and the description “*Aliquots of blank tap water samples from different regions and Haihe River water samples and those spiked with 50 ng/L of PFOS or PFOA were extracted with HLB cartridges (see the Supplementary Materials for detail). We then performed the displacement assay using the GC5A-6C•Fl reporter pair and obtained the concentrations of PFOS and PFOA from the established calibration curves. As shown in Supplementary Table 3, the experimental results showed that no PFOS or PFOA was detected in any of the blank tap water samples and Haihe River water samples. The standard addition recoveries of those water samples are in the range of 90.5~102.7%.*” in the first paragraph of Page 7.

Moreover, we added the description “*Water samples from the tap, Mati Lake and Haihe River were filtered through a nylon film (0.45 μ m) before experiments. The aliquots of blank tap water samples and those spiked with 50 ng/L of PFOS or PFOA were extracted with HLB cartridges. (6 cc, 150 mg; Waters Corp. Milford, U.S.A.). First, the cartridges were activated and conditioned with 5 mL methanol and 5 mL water. Second, the water samples (250 mL) were passed through the wet cartridges. Third, the columns rinsed with 5 mL sodium acetate buffer (25 mM) and 10 mL methanol, and the cartridges dried for 30 min by N_2 . Then, elution was performed with 2% ammonium hydroxide in methanol (7 mL). Finally, the eluate was then concentrated to 250 μ L for fluorescence displacement assay.*” in the Section 1.4 of the Supplementary Materials.

Thinking more broadly about the impact of their work on the field, whilst their system has obvious merits in the simultaneous detection and removal of PFOA, for any commercial applications one would suspect that the ligand would be too expensive and the removal of PFOA from the calixarene to regenerate the receptor-reporter complex is not described.

Response: Calixarenes are prepared through one-pot synthesis by using cheap reactants, such as 4-tert-butylphenol and formaldehyde. Calixarenes are also very easy to modify and thus were described as having “(almost) unlimited possibilities”

(Volker B, Angew. Chem. Int. Ed., 1995, 34, 713-745). Moreover, due to the high affinity and sensitivity to PFOS and PFOA, only very small amounts of the guanidinocalixarenes (1.14 $\mu\text{g/mL}$) were needed for each batch of detection application. To address the concern about regeneration, we performed the molecular recognition and assembly of GC5A-12C in DMSO. We have added the Supplementary Figs. 14 and 15, and the description “*The regeneration of the present supramolecular materials is considered to be feasible because supramolecular systems are environmentally sensitive, such as solvents. In DMSO, GC5A-12C neither formed amphiphilic aggregates indicated by very low scatter intensity (Supplementary Fig. 14), nor complexed with PFOS revealed by no change in chemical shifts of ^{19}F NMR spectra of PFOS (Supplementary Fig. 15). So we envisage that the regeneration of materials could be achieved by using routine purification methods in organic synthesis such as column chromatography.*” in Page 8.

One could also argue that using fluorescence is not as useful as optical methods for an in the field application.

Response: Analytical techniques based on fluorescence detection are very popular because of their low cost (Quang D, et al. Chem. Rev., 2010, 110, 6280-6301), easy to perform (Chen X, et al. Chem. Soc. Rev., 2010, 39, 2120-2135), high sensitivity (parts per billion/trillion), and real-time and non-invasive monitoring capability (Carter K P et al. Chem. Rev., 2014, 114, 4564-4601). To address the concern of the reviewer, we performed a real-time/on-site visual detection for measuring concentrations of PFOS and PFOA by using a smartphone with easy-to-access colour scanning application and added Fig. 6 in the revised manuscript.

Calixarenes are sometimes emissive in solution without any hosts. Did the authors check that their signals are coming from a host:guest complex and not just the host? I can't see this in the manuscript.

Response: According to the reviewer's advice, we additionally conducted the fluorescence experiments of the calixarene, dye and the host-guest complex. We have added the Supplementary Fig. 3, and the description “*Calixarenes are non-emissive in these experiments, which indicates the fluorescence signals come from the host-guest*

complex but not the host (Supplementary Fig. 3).” in the Section 2.3 of the Supplementary Materials.

No error bars on the data presented in Fig. 4, which is essential for a calibration graph.

Response: According to the reviewer’s kind reminder, we have added data error bars in the Fig. 4 and change the related description as “*Based on these results, we calculated the limit of detection (LOD) values as 21.4 ± 0.4 nM (11.3 ± 0.2 μ g/L) for PFOS and 26.4 ± 0.2 nM (10.9 ± 0.1 μ g/L) for PFOA by a 3σ /slope method.*” in Page 5 and “*The LOD values for PFOS were 30.9 ± 0.1 nM (16.6 ± 0.1 μ g/L) in the tap water and 120.8 ± 0.5 nM (65.0 ± 0.3 μ g/L) in Mati Lake. Meanwhile, the LOD values for PFOA were 39.1 ± 2.8 nM (16.2 ± 1.2 μ g/L) in the tap water and 120.3 ± 2.0 nM (49.8 ± 0.8 μ g/L) in Mati Lake.*” in Page 6.

Fig 5, do you mean chloride rather than chlorine? Again on Fig 5 what about metal cations as interfering species? One would assume that the concentration of Na⁺/K⁺ etc would be orders of magnitude more than PFOA?

Response: We used NaCl in Fig. 5. According to the reviewer’s suggestion, we have evaluated more metal cations as interfering species, such as NaCl, KCl, MgCl₂ and Na₂SO₄, as provided in the revised Fig. 5. We also added the description “*Salt concentrations are considered to be orders of magnitude higher than PFOS and PFOA in contaminated water. Even 1000-fold excess salts (NaCl, KCl and MgCl₂) resulted in much smaller fluorescence recovery of the GC5A-6C•Fl reporter pair than PFOS and PFOA (Supplementary Fig. 6).*” in Page 5 of the revised manuscript and the Supplementary Fig. 6 in the Section 2.5 of the Supplementary Materials.

In summary, whilst the science is interesting and could develop the thinking of others, calixarene and cyclodextrin receptors have already been reported. The authors have references wrong or missing from this list - the group of Hazendonk has published extensively on CD, the Baker group have used calixarene (not CD in the reference list) and the Allen group have used tripodal amides (Allen, J. Org. Chem., 2009, 74, 3706) to do the same supramolecular chemistry. In all these works the host:guest

complexes have been characterised in much greater detail, which leads this reviewer to err on the side of rejection due to the lack of novelty.

Response: We are very sorry for the mistake in describing the work of Baker. We have corrected the mistake and added the excellent works mentioned by the reviewer, which have been cited as Ref. 58, 66 and 75. The previous works reported by Allen and Hazendonk groups focused on structural characterization and dynamic properties of CD and PFOA/PFOS complexes. Baker and coworkers reported three types of calixarene for extract PFOA from water into an organic solvent. However, their binding affinities toward PFOA are about three orders of magnitude lower than our reported guanidinocalixarenes in this work. We designed artificial calixarene receptors with nanomolar strong and specific binding to PFOS and PFOA through the synergistic effect of electrostatic interactions, hydrogen bonds and van der Waals interactions. Stronger affinity will result in higher sensing sensitivity and absorption efficiency. Therefore, we achieved sensitive and selective fluorescence detection of PFOA and PFOS pollutants in real-world water samples. Moreover, through making full use of molecular recognition and self-assembly of the macrocyclic amphiphile, GC5A-12C, we incorporated efficiently absorption and remediation of PFOA and PFOS into the same system, which offered a promising new strategy for constructing integrated assemblies for the detection and remediation of water pollution.

The applicability of this approach to water remediation is still a long way off (note that this has no bearing on the scientific merits of the manuscript for the purposes of the review), but this reviewer encourages the authors to expand their characterisation data as this work does deserve to be published in some other journal.

Response: Thanks for the reviewer's constructive comments again. The reviewer has raised some critical and constructive questions and these comments are valuable for improving our manuscript. To make our approach more available to daily use, we developed a real-time/on-site visual detection mode for measuring concentrations of PFOS and PFOA by using a smartphone with easy-to-access colour scanning application (Fig. 6). We have added the corresponding discussion "***Real-time/on-site scanometric monitoring of PFOS and PFOA. To make the assay more available to***

daily use, the obvious fluorescence changes of GC5A-6C•Fl reporter pair with various concentrations of PFOS and PFOA were further applied as a real-time/on-site visual detection mode by using a smartphone with easy-to-access colour scanning application (app). The green colour intensities (G values) of the fluorescent images can be directly scanned from the app (Fig. 6). According to the G values, calibration curves can be set up with increasing PFOS and PFOA concentrations, respectively.” in Page 7 and “GC5A-6C•Fl (8.0/10.0 μ M) reporter pair in HEPES buffer was mixed up with various concentrations of PFOS and PFOA for detection. The solution (2.0 mL) of each group was added into a tube of 2.5 mL, followed by exciting with a 254 nm UV lamp. Then, colour change could be taken by a smartphone and the images were handled with a custom developed colour scanning application from Apple Store (World of Color, Maarten Zonneveld). RGB intensities were displayed on the screen and G value was extracted for determination of PFOS and PFOA.” in the Section 1.5 of the Supplementary Materials.

We have done our best to revise our manuscript according to the reviewer's advices, and therefore, the quality of the manuscript has been much improved. We believe that the revised manuscript merits publication in *Nature Communications*.

Reply to Reviewer 2

Comments:

In this manuscript, authors have developed a new strategy for the direct detection and rapid removal from water of two persistent organic pollutants (PFOS and PFOA) that are currently causing increasing environmental and health concern. The detection is based on a fluorescent indicator-displacement assay that relies on the strong affinity (in the nanomolar range) of guanidinocalix[5]arenes GC5A-6C and GC5A-12C towards these perfluorocarbon derivatives. The removal procedure takes advantage of the amphiphilic nature of GC5A-12C and the magnetic properties of iron oxide nanoparticles that, combined in the presence of PEG-12, yield hybrid nanoparticles able to absorb, magnetically separate and remove, after simple filtration, the target analytes from water samples.

The manuscript is well written and relevant literature has been thoroughly reviewed. Experiments performed and methods used are clearly described. Discussion and conclusions are sound and backed up by solid experimental evidence (DFT calculations, ^{19}F NMR studies, DLS measurements and UPLC-ESI-MS/MS analyses). This is a nice piece of work where molecular recognition and self-assembly properties of amphiphilic macrocycles have been cleverly combined to address an issue of current concern related to water pollution and, as a result, bioaccumulation problems and risk to human health. The detection limits reported are promising and the idea of using a magnetic separation technique is interesting. The findings by Guo and colleagues are of a high standard and will, in my opinion, provide the input for new research to be carried out in the field of environmental and separation science.

Response: We highly appreciate the reviewer's positive and constructive comments.

Publication of this article in Nature Communications is recommended following minor revision/additions.

Revision/additions to be addressed:

1. Information concerning the regeneration of the MNP@GC5A-12C nanoparticles for subsequent reuse is lacking.

Response: To address the concern about regeneration, we performed the molecular recognition and assembly of GC5A-12C in DMSO. We have added the Supplementary Figs. 14 and 15, and the description “*The regeneration of the present supramolecular materials is considered to be feasible because supramolecular systems are environmentally sensitive, such as solvents. In DMSO, GC5A-12C neither formed amphiphilic aggregates indicated by very low scatter intensity (Supplementary Fig. 14), nor complexed with PFOS revealed by no change in chemical shifts of ¹⁹F NMR spectra of PFOS (Supplementary Fig. 15). So we envisage that the regeneration of materials could be achieved by using routine purification methods in organic synthesis such as column chromatography.*” in Page 8.

2. The choice of using HEPES buffered solutions probably deserves a comment.

Response: HEPES has a pK_a value of 7.5, which is suitable for a neutral buffer. Moreover, HEPES is highly soluble, chemically stable, with very low visible and ultraviolet light absorbance, and easy to prepare. Consequently, HEPES buffered solutions are widely used in various areas, including detection and absorption of pollutants from wastewater (Mines P D, et al. J. Mater. Chem. A, 2016, 4, 632-639; Chow C F, et al. Chem. Sci., 2017, 8, 3812-3820 and Cho Y, et al. Analyst, 2010, 135, 1551-1555).

3. Has selectivity towards simple perfluorocarbons been tested?

Response: According to the reviewer’s suggestion, we have evaluated a simple perfluorocarbon interfering specie, perfluorohexane, as provided in the revised Fig. 5.

4. The type of film used for the filtration of the supernatant obtained after the magnetic absorption should be specified (line 191)

Response: We have added the description “*Subsequently, the supernatant was filtered through a mixed cellulose esters film (Millipore, 0.025 μm) and collected for UV-Vis experiments*” in Page 8.

5. Replace “a family” with “families” (line 61)

Response: We replaced “a family” in the revised manuscript with “*families*”.

6. Reference 72 (line 65) deals with calix[4]arenes rather than beta-cyclodextrins!

Response: We are very sorry for this mistake and the reference has been properly cited as Ref. 58 in the revised manuscript.

7. Authors might want to consider introducing the following publication among the references cited in the introduction: doi: 10.2903/j.efsa.2018.5194

Response: We have noted the excellent review papers mentioned by the reviewer (doi: 10.2903/j.efsa.2018.5194), which has been cited as Ref. 6 in our manuscript.

8. The caption to Fig 2 in the SI needs attention. I also wondered whether there was any specific reason for the poor phasing of the ^{19}F NMR traces shown in Fig 2 of the SI.

Response: Due to the bad phasing and peak shape of the GC5A-6C•PFOA complex in D_2O , we added NMR spectra of the CD•PFOA complex in D_2O for comparison. We have deleted Supplementary Fig. 2 because the further performed ^{19}F NMR spectra of the GC5A-6C•PFOA complex in $\text{MeOH-}d_4$ have good phasing and peak shape.

9. Poor trace phasing is also seen in Fig 3 of the SI.

Response: We have further performed ^{19}F NMR in $\text{MeOH-}d_4$ due to the low solubility of the GC5A-6C•PFOA complex in D_2O . The updated results have good phasing and can validate the binding between GC5A-6C and PFOA (see revised Supplementary Fig. 2), although smaller upfield shifts due to lower polarizability of $\text{MeOH-}d_4$ than D_2O .

Reply to Reviewer 3

Comments:

The submission by Zheng and co-workers reports a new approach to binding and removing PFOS and PFOA from water. These are highly important contaminants that are of increasing interest in the last ten years. The idea of using a general hydrophobic sorbent to reduce PFOS/PFOA levels in water is well established, and could use a better coverage in the literature citations in this work. The two main new ideas in this work are the use of a macrocyclic host for binding the fluorinated pollutants, and combining the ability to detect with the ability to remove the pollutants in the same molecular framework.

The writing is very clear and the science is well presented. The binding constants, selectivity, mode of action, and ability to remove pollutants from water are all well supported by the data presented.

In order for the work in this area to be suitable for Nature Communications and attract a broad readership, it needs both a high level of scientific innovation and a significant step toward real-world application. The innovation is indeed very high. The studies with lake water are a nice addition and they bring this quite close to real-world impact.

Response: We highly appreciate the reviewer's positive and constructive comments.

The sorbent studies fall a bit short, mainly because the UPLC-MS method that is used to quantify PFOS/PFOA after removal are not sensitive enough to determine exactly how complete the removal of PFOS is. This ability is central to the future utility of this method, and the analysis must be possible because it is done routinely in environmental analysis labs. I ask that the same experiments be reported with accurate determination of PFOS/PFOA concentrations down at least to the government standard of 70 ng/L.

Response: According to the reviewer's advice, we optimized the measurement procedures. After collecting the residuals, we pre-concentrated them from 2.0 mL to

100 μ L before accurately determining PFOS/PFOA concentrations by means of UPLC-ESI-MS/MS. We have updated Fig. 8b and added the description “*After the removal procedure, the solutions were pre-concentrated to accurately determine PFOS and PFOA concentrations by the UPLC-ESI-MS/MS. There were only $(0.43 \pm 0.07)\%$ residual PFOS and $(1.53 \pm 0.04)\%$ residual PFOA, respectively (Fig. 8b).*” in Page 8.

The other possible shortcoming is about selectivity, which is central to innovation and to the application. This selectivity needs to be established experimentally using real highly contaminated waters. There is a significant chance that the macrocycle will find other hydrophobic pollutants to bind preferentially over PFOS/PFOA, and this would remove the proposed advantage of the macrocyclic system, and more importantly would also decrease the innovation aspect of the work relative to nonspecific sorbents that have already been reported. If this can be established it could spark others to try their own combinations of specific macrocyclic/MNP sorbents for different species.

Response: According to the reviewer’s suggestion, we have collected highly contaminated water samples and added the description “*The waste water samples were provided by the manufacturing facility located in Cangzhou, Hebei province, China.*” in the Method section. We have also added the description “*Considering that other unknown pollutants in real highly contaminated water might interfere with the detection selectivity of GC5A-6C, we collected the waste water samples from the manufacturing facility (Cangzhou, Hebei province, China), and freeze-dried these highly contaminated water samples. The addition of the obtained powder with mass concentration more than 10 times higher than PFOS and PFOA also caused no significant enhancement in the fluorescence intensity, which validated GC5A-6C could bind preferentially to PFOS and PFOA over other pollutants in the real world. Moreover, the selectivity to various interferences in Fig. 5 were established in the real highly contaminated water (Supplementary Fig. 7).*” in Page 6. In summary, the designed guanidinocalixarenes have good selectivity toward PFOA and PFOS benefiting from the synergistic recognition, even in real highly contaminated water.

The advantage of selective recognition relative to nonspecific sorbents could inspire other researchers for using specific macrocyclic receptors in treating other types of pollutants in waste water.

REVIEWERS' COMMENTS:

Reviewer #1 (Remarks to the Author):

I was pleasantly surprised by the nature and scale of the revisions the authors of this work have conducted. The manuscript now reads very well and all my criticisms have been ameliorated. I have only minor comments that need attention in the manuscript:

1. The authors report the binding constants for PFOA and PFOS but with little context. Please add (maybe in a table) the reported binding constants from the literature; this adds more weight to their novelty.

2. Does octanoic acid (or any other carboxylic acid) bind?

3. The smartphone idea is very good. However how do the authors propose to address the issues of different phones having different cameras and lighting bias? I fully see the advantages of this methodology for in-field testing, but it's not as simple as it seems. For more information I suggest the authors see:

<https://pubs.rsc.org/en/content/articlelanding/2016/ay/c6ay01575a#!divAbstract>. At the very least the authors need to be specific in the phones tested and add a rider that it may not work exactly that way on all phones. I was a bit disappointed to see in the text that the software was "custom designed". This limits the reproducibility of the authors' work if this is not available.

I'd finally like to congratulate the authors on the effort put in to improve their manuscript.

Reviewer #2 (Remarks to the Author):

The authors have adequately dealt with the concerns raised by the reviewers. The additions provided have further improved the standard of the manuscript which, in my opinion, deserves to be published in Nature Communications. Some very minor suggestions remain, however, to be addressed.

- Authors are encouraged to refer to tetradeuteromethanol as "CD₃OD" rather than "MeOH-d₄" (in line 108 of the main text and in the caption to the supplementary Fig 2).
- Without questioning the formation of a GC5A-6C/PFOA endo-cavity complex, comments on the complexation induced shifts (line 111) should probably just focus on the alpha-CF₂ resonance. According to the supplementary Fig 2, this is the only signal that undergoes a noticeable downfield shift in CD₃OD. In addition, I have just noticed that the lettering of the groups of PFOA (in the supplementary Fig 2) does not follow the normal order of the Greek alphabet.
- A description of what the picture in Fig 6 shows should be provided in the caption. Furthermore, information on the type of "tube" used for the G-value calibration should also be added in section 1.5 of the Supplementary Materials.
- The last three sentences of page 8 (lines 228–234) could benefit from some linguistic improvement.
- Beta-cyclodextrin should be deleted from the list of chemicals quoted in the "Methods" section.

Reviewer #3 (Remarks to the Author):

The authors have addressed my concerns completely. The choice to do the studies with other 'real' contaminants at very high levels using lyophilized powder was a good choice. That, together with the very selective responses now shown in Figure 5, combine to show that the system does work in real world samples. This validates the scientific leap forward that was proposed, which is that macrocycles will give PFOA and PFOS selectivity that is not available from generalized hydrophobic

sorbents. The improvements to the citations of other relevant literature (which were suggested by referees 1 and 2) also make the position of this innovative set of results much more clear. The development of a smart phone app is nice, but ultimately not necessary. I now find this worthy of publication in Nature Commun on the grounds of both novelty and importance.

Reply to Referee 1

Comments:

I was pleasantly surprised by the nature and scale of the revisions the authors of this work have conducted. The manuscript now reads very well and all my criticisms have been ameliorated. I have only minor comments that needs attention in the manuscript.

Response: We highly appreciate the reviewer's positive comments.

1. The authors report the binding constants for PFOA and PFOS but with little context. Please add (maybe in a table) the reported binding constants from the literature; this adds more weight to their novelty.

Response: According to the reviewer's advice, we have added the description "By employing GC5A-6C•Fl as the reporter pair ($K_a = 5.0 \times 10^6 M^{-1}$), we obtained binding affinities of $(3.5 \pm 1.0) \times 10^7 M^{-1}$ for PFOS (Fig. 3b) and $(1.7 \pm 0.3) \times 10^7 M^{-1}$ for PFOA (Fig. 3c). The binding affinities are about three orders of magnitude higher than those of previously reported supramolecular hosts toward PFOS, which are around $10^4 M^{-1}$ (Supplementary Table 3)." in the last paragraph of Page 4 of the revised manuscript and added Supplementary Table 3 in the revised Supplementary Information.

2. Does octanoic acid (or any other carboxylic acid) bind?

Response: We have evaluated the binding constants of octanoic acid and octanesulfonic acid, as the description "For comparison, we determined that octanesulfonic acid and octanoic acid (C-H chain surfactants) exhibit affinities of $(6.0 \pm 1.1) \times 10^4 M^{-1}$ (Supplementary Fig. 4) and $(7.6 \pm 0.8) \times 10^4 M^{-1}$ (Supplementary Fig. 5) with GC5A-6C, respectively." in the last paragraph of Page 5.

3. The smartphone idea is very good. However how do the authors propose to address the issues of different phones having different cameras and lighting bias? I fully see the advantages of this methodology for in-field testing, but its not as simple as at seems. For more information I suggest the authors see: <https://pubs.rsc.org/en/content/articlelanding/2016/ay/c6ay01575a#!divAbstract>. At the very least the authors need to be specific in the phones tested and add a rider that

it may not work exactly that way on all phones. I was a bit disappointed to see in the text that the software was "custom designed". This limits the reproducibility of the authors work if this is not available.

Response: We are very sorry for the misleading description "custom designed". The colour scanning application can be directly downloaded from Apple store. According to the reviewer's suggestion, we have revised the description "*Then, colour change could be taken by an iPhone 7 and the images were handled with a colour scanning application from Apple Store (World of Color, Maarten Zonneveld). For other models of smartphones, the present calibration curves for iPhone 7 may not work exactly and new calibration curves may need be set up.*" in Supplementary Note 3.

I'd finally like to congratulate the authors on the effort put in to improve their manuscript.

Response: We highly appreciate the reviewer's positive comments again.

Reply to Referee 2

Comments:

The authors have adequately dealt with the concerns raised by the reviewers. The additions provided have further improved the standard of the manuscript which, in my opinion, deserves to be published in Nature Communications.

Response: We highly appreciate the reviewer's constructive comments.

Some very minor suggestions remain, however, to be addressed.

1. Authors are encouraged to refer to tetradeuteromethanol as "CD₃OD" rather than "MeOH-d₄" (in line 108 of the main text and in the caption to the supplementary Fig 2).

Response: We replaced "MeOH-d₄" with "CD₃OD" in the revised manuscript and the caption to the Supplementary Fig. 2.

2. Without questioning the formation of a GC5A-6C/PFOA endo-cavity complex, comments on the complexation induced shifts (line 111) should probably just focus on the alpha-CF₂ resonance. According to the supplementary Fig 2, this is the only

signal that undergoes a noticeable downfield shift in CD3OD. In addition, I have just noticed that the lettering of the groups of PFOA (in the supplementary Fig 2) does not follow the normal order of the Greek alphabet.

Response: According to the reviewer's advice, we added the description "*The significant upfield shift in the fluorine nuclei link to C_α was observed (Supplementary Fig. 2), which may be caused by the intermolecular N–H···F hydrogen bonds between guanidinium groups of GC5A-6C and fluorine atoms link to C_α of PFOA when PFOA was immersed into the cavity of GC5A-6C (Supplementary Fig. 1).*" in Page 4. Moreover, we have revised the lettering of the groups of PFOA follow the normal order of the Greek alphabet in the Supplementary Fig. 2.

3. A description of what the picture in Fig 6 shows should be provided in the caption. Furthermore, information on the type of "tube" used for the G-value calibration should also be added in section 1.5 of the Supplementary Materials.

Response: According to the reviewer's suggestion, we have revised the caption of Fig. 6 and added the description "*The solution (2.0 mL) of each group was added into a polypropylene centrifuge tube (BBI Life Sciences, 2.5 mL), followed by exciting with a 254 nm UV lamp.*" in the Supplementary Note 3.

4. The last three sentences of page 8 (lines 228–234) could benefit from some linguistic improvement.

Response: We appreciate the reviewer's constructive advices. We have revised the description "*The regeneration of the present supramolecular material is considered to be feasible owing to its response to specific organic solvent. In DMSO, GC5A-12C neither formed amphiphilic aggregates indicated by very low scattering intensity (Supplementary Fig. 14), nor complexed with PFOS revealed by no change in chemical shifts of ¹⁹F NMR spectra of PFOS (Supplementary Fig. 15). Therefore, we envisage that regeneration of the supramolecular material could be achieved by using routine purification methods in organic synthesis such as column chromatography.*" in Page 8.

5. Beta-cyclodextrin should be deleted from the list of chemicals quoted in the "Methods" section.

Response: We deleted “ β -cyclodextrin (β -CD)” in the “Methods” section.

Reply to Referee 3

Comments:

The authors have addressed my concerns completely. The choice to do the studies with other ‘real’ contaminants at very high levels using lyophilized powder was a good choice. That, together with the very selective responses now shown in Figure 5, combine to show that the system does work in real world samples. This validates the scientific leap forward that was proposed, which is that macrocycles will give PFOA and PFOS selectivity that is not available from generalized hydrophobic sorbents. The improvements to the citations of other relevant literature (which were suggested by referees 1 and 2) also make the position of this innovative set of results much more clear. The development of a smart phone app is nice, but ultimately not necessary. I now find this worthy of publication in Nature Commun on the grounds of both novelty and importance.

Response: We highly appreciate the reviewer’s positive comments.